# Pharmacogenetic Sex-Specific Effects of Methotrexate Response in Patients with Rheumatoid Arthritis

**DOI:** 10.3390/pharmaceutics15061661

**Published:** 2023-06-05

**Authors:** Francisco C. Ceballos, Eugenio Chamizo-Carmona, Carmen Mata-Martín, Carmen Carrasco-Cubero, Juan J. Aznar-Sánchez, Raúl Veroz-González, Sara Rojas-Herrera, Pedro Dorado, Adrián LLerena

**Affiliations:** 1Health Institute Carlos III (ISCIII), 15706 Santiago de Compostela, Spain; 2Department of Rheumatology, Hospital de Mérida, 06800 Mérida, Spain; 3MEPER Group-Clinical and Translational Research in Pharmacogenetics and Personalized Medicine, Biosanitary Research Institute of Extremadura (INUBE), 06080 Badajoz, Spain; 4CICAB Clinical Research Center, Pharmacogenetics and Personalized Medicine Unit, Hospital Universitario de Badajoz, 06080 Badajoz, Spain; 5Department of Rheumatology, Hospital Universitario de Badajoz, 06080 Badajoz, Spain; 6Faculty of Medicine, Universidad de Extremadura, 06071 Badajoz, Spain

**Keywords:** methotrexate, pharmacogenetics, rheumatoid arthritis, response, efficacy, adverse effects

## Abstract

Methotrexate (MTX) is a commonly used drug for the treatment of rheumatoid arthritis (RA), but its effectiveness can vary greatly among patients. Pharmacogenetics, the study of how genetic variations can affect drug response, has the potential to improve the personalized treatment of RA by identifying genetic markers that can predict a patient’s response to MTX. However, the field of MTX pharmacogenetics is still in its early stages and there is a lack of consistency among studies. This study aimed to identify genetic markers associated with MTX efficacy and toxicity in a large sample of RA patients, and to investigate the role of clinical covariates and sex-specific effects. Our results have identified an association of *ITPA* rs1127354 and *ABCB1* rs1045642 with response to MTX, polymorphisms of *FPGS* rs1544105, *GGH* rs1800909, and *MTHFR* genes with disease remission, *GGH* rs1800909 and *MTHFR* rs1801131 polymorphisms with all adverse events, and *ADA* rs244076 and *MTHFR* rs1801131 and rs1801133, However, clinical covariates were more important factors to consider when building predictive models. These findings highlight the potential of pharmacogenetics to improve personalized treatment of RA, but also emphasize the need for further research to fully understand the complex mechanisms involved.

## 1. Introduction

Rheumatoid arthritis (RA) is a chronic autoimmune disease that causes inflammation of the synovial tissue, leading to destruction of bones and cartilage [1]. It affects between 0.5% and 1.0% of the population, with a higher prevalence in women and urban areas in the northern hemisphere [2]. The European League Against Rheumatism (EULAR) has been using, since the 1980s, methotrexate (MTX), a conventional synthetic Disease-modifying antirheumatic drug (DMARD), as the first-line drug for RA treatment. However, MTX has a significant inter-patient variability, with 30–50% of patients failing to achieve remission, and up to 30% experiencing adverse effects that exclude them from treatment due to toxicity [3,4]. MTX is also used in the treatment of other autoimmune diseases (such as psoriasis, 10 to 25 mg/week) and certain types of cancer (such as lymphoblastic leukemia, 200 mg/m^2^ intravenous over two hours up to high doses of 1–3 g/m^2^ as a 24 h continuous infusion), usually a somewhat higher dose than for RA (7.5 to 20 mg/week) [5].

MTX has been studied to explain the variability in clinical response to the drug [4]. To date, over 120 SNPs in 34 genes have been identified as potentially contributing to treatment response [6,7,8]. However, it is important to replicate these results and continue searching for new genes associated with the response to MTX therapy [9,10,11,12]. Studies on pharmacogenetic risk factors for MTX toxicity and efficacy have produced conflicting data, potentially due to limited sample size, ancestry heterogeneity, and lack of accounting for comorbidities and other confounding factors.

Inconsistent results have been found for more than 20 SNPs in 15 genes, highlighting the need for further research (Table 1). For example, the interaction between MTX and the *MTHFR* (5,10-methylenetetrahydrofolate reductase) gene within the folate pathway is unclear. *MTHFR* is one of the most studied genes in this context, but its relationship with treatment response or toxicity is still not clear [7,13,14,15]. Other well-studied genes involved in the MTX pathways, such as those belonging to the ABC (multi-drug resistance protein) family pumps, RFC1 (replication factor C subunit 1), ATIC (5-aminoimidazole-4-carboxamide ribonucleotide formyltransferase/IMP cyclohydrolase), FPGS (folylpolyglutamate synthase), GGH (gamma-glutamyl hydrolase), and TYMS (thymidylate synthetase), have also shown conflicting results [6,7,16]. Additionally, although differences between men and women have been found to be significant in the genetics of RA [17] and in pharmacogenetics in general [18], to the best of our knowledge, no studies have focused specifically on the biological differences between the genders with respect to these genes.

The aim of this study was to analyze the association between different polymorphisms of genes encoding proteins involved in the metabolism and transport of MTX, and the efficacy and safety in a group of patients with RA treated with MTX. In addition, the biological differences between men and women with respect to these genes will be evaluated.

## 2. Materials and Methods

### 2.1. Participant Recruitment

For this study, we used the Merida cohort, which consists of RA patients recruited since March 1990 and followed up in monographic medical consultations followed between 2012 and 2015 at the Hospital General de Mérida. These patients were recruited into a prospective observational cohort study using the following inclusion and exclusion criteria. Inclusion criteria: patients over the age of 18 years who were followed between 2012 and 2015, diagnosed with RA according to American College Rheumatology criteria [28], protocolized follow-up in the RA monographic consultation (this information is on the patient’s evaluation sheet), receiving, or has ever received, treatment with MTX monotherapy. During monotherapy, patients could also take concomitant nonsteroidal anti-inflammatory drugs, oral prednisone, and folic acid, but not other DMARDs from baseline. Exclusion criteria: high probability of patient loss, inability to provide accurate data, and patients enrolled in clinical trials. Furthermore, patients who had received biologics, cyclophosphamide, or a combination of MTX with other DMARDs prior to monotherapy were excluded.

Informed consent was obtained from all RA patients, and the study was approved by the Bioethics and Biosafety Commission of the University of Extremadura (No. 42/2011).

MTX was administered under a rapid escalation pattern, initially administered orally in a single dose up to 10 mg/day, in two doses starting at 10 mg/day. Patients were switched to the subcutaneous route if they did not tolerate oral administration well, the amount of 15 mg per day was exceeded, or the desired efficacy was not achieved.

### 2.2. DNA Extraction and Genotyping

Eight polymorphisms in seven genes encoding proteins involved in MTX metabolism (pharmacokinetics and pharmacodynamics) and MTX toxicity were selected for genotyping in our patients according to a literature review (Figure 1). The analyzed polymorphisms were:-MTX extraction pumps: *ABCB1* rs1045642 (3435C>T).-MTX polyglutamate formation: *GGH* rs1800909 (T16C) and *FPGS* rs1544105 (G2782A).-Folate cycle: *MTHFR* rs1801131 (C677T) and *rs1801133* (A1298C).-Adenosine pathway: *AMPD1* rs17602729 (34C>T), *ITPA* rs1127354 (94C>A), and *ADA* rs244076 (534A>G).

Genomic DNA was obtained from blood samples storage in EDTA (Vacutainer, BD Diagnostics, Franklin Lakes, NJ, USA) at −20 °C. DNA isolation was obtained using a Gentra Puregene DNA Isolation Kit (Gentra Systems, Minneapolis, MN, USA) according to the protocol provided by the manufacturer. The eight polymorphisms were genotyped by employing real-time polymerase chain reaction (RT-PCR) using TaqMan probes in a Fast 7300 Real-Time platform (Applied Biosystems, Forster City, CA, USA)

### 2.3. Variables of Interest

Two indicators of MTX efficacy were obtained based on the Disease Activity Score (DAS) 28 with C-reactive protein (DAS28-CRP) levels: response to MTX (DAS28 < 3.2) [29] and RA remission (DAS28 < 2.6). The DAS is a scoring system to assess the activity of rheumatoid disease, having been recommended by the EULAR for this purpose both in clinical studies and in daily clinical practice. The DAS index for measuring disease activity was developed during the 1980s [30] and was published by Van der Heijde et al. (1990) [31]. DAS uses 44 joints in its count. The DAS index combines information regarding the number of swollen, tender, and acute phase reactant joints, and a global measure of health status. DAS28 is a simplified version using only 28 joints for counting and was published by Prevoo et al. (1995) [32]. Therefore, DAS28-CRP is a variant of DAS28 that substitutes the erythrocyte sedimentation rate for the C-reactive protein count and takes 28 specific joints.

To establish response criteria, patients had to have received MTX monotherapy at the highest tolerated dose for at least 6 months and responders who achieved and maintained a DAS28-CRP < 2.6 were in clinical remission, regardless of whether they continued MTX monotherapy or no treatment.

The events included as adverse events were hematological affection (anemia: Hb < 9 gm/dL; leukopenia: leukocytes < 4000/μL; thrombocytopenia: platelet count < 150,000 mcL), hepatic affection as transaminase elevation (GOT and GPT) two times higher than the superior normality limit (6 < GOT < 30 UI/L; 6 < GPT < 35 UI/L), gastrointestinal affection (nausea, vomiting, diarrhea, abdominal pain, anorexia), and other general adverse events like fatigue, arthralgia, fever, and myalgia.

Forty-four cofactors and comorbidities were obtained to complete the statistical model (Table 2). Demographic and clinical variables with more than 5% of cases were added to the analysis. Included confounding variables can be organized into different groups:

Demographic variables (n = 8): age, sex at birth, educational level, dedication, consumption of alcohol, tobacco, coffee, and tea.

Comorbidities (n = 17): any comorbidities (yes or no), number of risk factors, high blood pressure, diabetes mellitus, hyperlipidemia, obesity, ischemic heart disease, any cardiovascular pathology, vascular incidents, thyroid disease, renal impairment, osteoporosis, tuberculosis, baldness, hypertransaminasemia, and depression.

Disease manifestations (n = 6): DAS28 basal levels (before treatment), extra-articular manifestations, radiological erosions, Sjogren syndrome, carpal tunnel syndrome, pulmonary affection.

Pharmacological variables (n = 10): use of DMARDs previously, number of previous DMARDs, time between diagnosis and treatment with DMARD, time between first symptoms and treatment with DMARD, initial doses of prednisone, doses of folic acid, maximum doses of MTX, maximum tolerated doses of MTX, time between diagnosis and treatment with MTX, MTX monotherapy duration.

Antibodies (n = 3): rheumatic factor, antipeptide citrullinated, antinuclear antibodies.

### 2.4. Statistical Analysis

Multivariable logistic models were used to assess the association between genetic variants and the dependent variables: response to MTX RA remission and adverse effects, as dichotomous variables. The 44 confounding variables gathered in this study were added to each of the linear models. In this first approach, we were not interested in obtaining the best explanatory model but rather in removing any possible confounding effect these variables may have on the dependent variables. For each genotyped SNP, we tested four different models of inheritance: dominant, recessive, codominant, and overdominant.

Once the different associations between genetic variants and the dependent variables were assessed, we used multivariable lasso logistic regression and receiver operating characteristic (ROC) curves to weight the contribution of the significant genetic variants in a prediction model. Lasso logistic regression was developed by randomly reallocating individuals into two groups, training, and test, with equal sizes. K-fold cross-validation was used with 150 folds.

All statistical analyses were performed using R v4.2. Significance was assessed using the False Discovery Rate, and the STROBE (strengthening the reporting of observational studies in epidemiology) methodology was followed [33].

## 3. Results

### 3.1. Clinical Characteristics

The aim of this study was to assess the effect of different genetic polymorphisms on the response to MTX in a group of RA patients treated with this drug. Table 2 shows details about the clinical characteristics of cases and controls.

The cohort was composed of 301 patients (67.1% women) with a median age of 49 years. Figure 2, Figure 3 and Figure 4 show the distribution of response to MTX, RA remission, and the different adverse events for each SNP, respectively.

Due to the lack of cases of hematological adverse events and hepatic adverse events (4.6% and 8.9% of cases, respectively; Table 3), the analysis was developed for the total number of adverse events (50.5%) and for gastrointestinal adverse events (28.9%). This is a retrospective study with no previous sample size calculation.

### 3.2. Allele Frequencies

Figure 1 shows the allele frequencies and presents a relative comparison between genotypes in a pie chart. In general, the study has a good representation of both alleles for each SNP, but this is not the case for *ITPA* and *AMPD1*, where very few alternative alleles are present (9.4% and 11.5%, respectively), and in most cases, they are in heterozygous form. As a result, we have a small representation of homozygous genotypes for these SNPs (seven individuals for *ITPA* and two individuals for *AMPD1*), which prevents the testing of certain models of inheritance.

### 3.3. Association between Genetic Variants and MTX Efficacy

The efficacy of the MTX treatment was measured through two variables: response to MTX (DAS28 < 3.2) and AR remission (DAS28 < 2.6).

In Figure 3, forest plots of the different analyses are shown, all the 44 confounding variables were added to the multivariable linear methods. This figure shows that *ITPA* rs1127354 homozygous C/C genotype is associated with having a higher probability of being a responder to MTX (logOR = 2.7; 95% CI = 0.8, 4.6) (Figure 3). Thus, chances of being a responder decrease when comparing codominant inheritance C/A vs. C/C or overdominant C/A vs. A/A + C/C and increase in a dominant model (C/C vs. C/A + A/A). *ITPA* effect was not significant when AR remission was considered.

**Figure 3 pharmaceutics-15-01661-f003:**
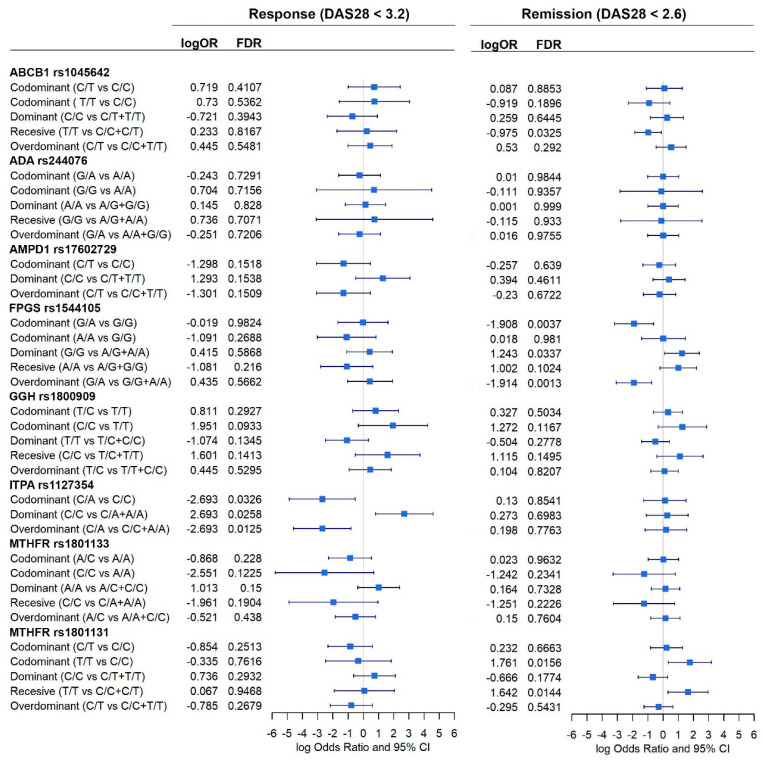
Effect of each gene, according to different inheritance models, on AR remission (DAS28 < 2.6) and response to MTX (DAS28 < 3.2). Outcomes are present in a forest plot showing effect estimates (logOdds ratio) with their 95% confident intervals.

*FPGS* rs1544105 and *MTHFR* rs1801131 polymorphisms showed association with AR remission. Dominant inheritance of *FPGS* homozygous G/G genotype is also associated with a higher probability of AR remission under a codominant (logOR = −1.9; 95% CI = −3.2, −0.6) and dominant models (logOR = 1.3; 95% CI = 0.1, 2.4). In contrast, *FPGS* heterozygote G/A genotype decreases the probability of AR remission under an overdominant inheritance (OR = −1.9; 95% CI = −3.1– 0.75). Finally, *MTHFR* rs1801131 T/T homozygous genotype is associated with a higher probability of AR remission in a recessive (logOR = 1.6; 95% CI = 0.3, 3.0) or codominant inheritance (logOR = 1.8; 95% CI = 0.3, 3.2) (Figure 3).

In order to analyze the sex-specific pharmacogenetic effects, the interaction between sex at birth and each genotype was assessed. For those SNPs showing a significant interaction effect with sex at birth, a separate analysis by sex was developed. In Figure 4, panels a and b, it is possible to find some exploratory figures with *GGH*, *FPGS* and *ABCB1* genotypes, response to MTX and AR remission distributions by sex, while panels c and d show the sex-specific effects in response to MTX and AR remission, for different genes and SNPs (Figure 4).

**Figure 4 pharmaceutics-15-01661-f004:**
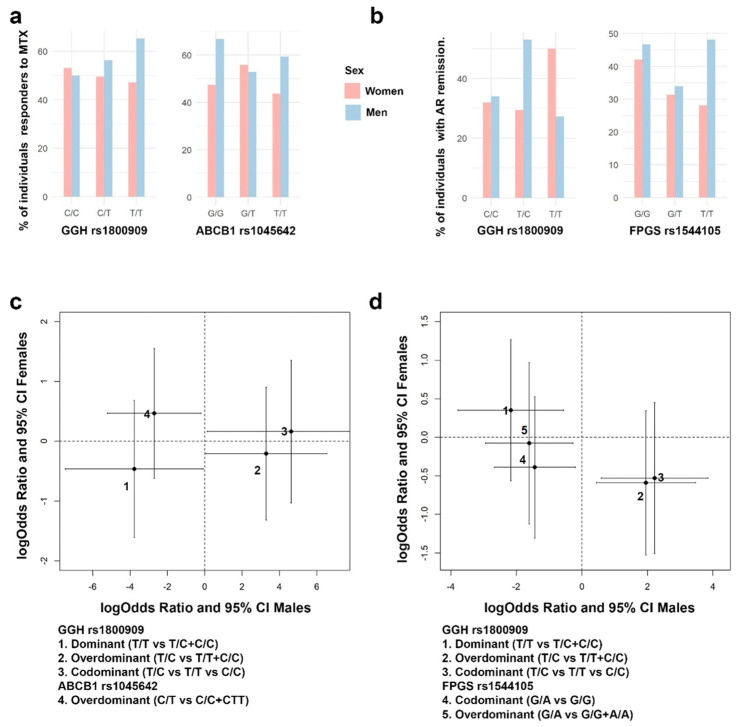
Sex-specific effects in AR remission (DAS28 < 2.6) and response to MTX (DAS28 < 3.2). (**a**) Bar plots with the percentage of responders to MTX (DAS28 < 3.2) in each SNP genotype for *GGH* rs1800909 and *ABCB1* rs1045642. (**b**) Bar plots with the percentage of individuals with AR remission (DAS28 < 2.6) in each SNP genotype for *GGH* rs1800909 and *FPGS* rs1544105. (**c**) *GGH* rs1800909 and ABCB1 rs17602729 effect estimates (logOdds ratio) over response to MTX, and 95% CI, for men (*x* axis) and women (*y* axis) are shown. (**d**) *GGH* rs1800909 and *FPGS* rs1544105 effect estimates (logOdds ratio) over response to MTX, and 95% CI, for men (*x* axis) and women (*y* axis) are shown.

Regarding response to MTX (Figure 4c), the effects were male specific, no significant effects were found in women. Thus, *GGH* rs1800909 heterozygous T/C was found to be associated with an increase prevalence of responders to MTX under an overdominant (logOR = 3.3; 95% CI = 0.1, 6.5) and codominant (logOR = 4.6; 95% CI = 0.1, 9.1) inheritance model. On the contrary, the dominant homozygous T/T decreased the prevalence of responders (logOR = −3.8; 95% CI = −7.5, −0.1). Response to MTX is also associated negatively with the overdominant model of *ABCB1* rs1045642 C/T (logOR = −2.7; 95% CI = −5.2, −0.2).

The AR remission (Figure 4d) showed the same sex-specific effects. Specifically, for the *GGH* rs1800909 SNP the pattern is exactly the same but with different effects sizes: positive association with overdominant (logOR = 2.0; 95% CI = 0.4, 3.5) and codominant (logOR = 2.2; 95% CI = 0.6, 3.9) inherited models and negative with a dominant model T/T (logOR = −2.2; 95% CI = −3.8, −0.6). However, no association was found with *ABCB1* rs1045642 but with FPGS rs1544105. *FPGS* heterozygote G/A decreases the probability of AR remission under a codominant (logOR = −1.6; 95% CI = −2.9, −0.3) and overdominant (logOR = −1.4; 95% CI = −2.7, −0.2).

### 3.4. Association between Genetic Variants and MTX Adverse Events

Due to our sample size and the number of adverse events, analyses were only possible using the total number of adverse events and, specifically, the gastrointestinal adverse events, as can be seen in Figure 5. In Figure 6, forest plots of the different analyses are shown, all 44 confounding variables were added to the multivariable linear methods.

Significant association between genes and adverse events were found (Figure 6).

When all events are considered, *ADA* rs244076, *GGH* rs1800909, and *MTHFR* rs1801131 polymorphisms were shown to affect the prevalence of adverse events. Homozygous C/C *GGH* increased the prevalence of adverse events under a recessive model (logOR = 1.2; 95% CI = 0.03, 2.3). Heterozygous C/T *MTHFR* rs1801131 genotype also increased this prevalence under codominant (logOR = 1.17; 95% CI = 0.4, 1.9) and overdominant (logOR = 1.19; 95% CI = 0.4, 1.9) inheritance models. On the contrary, homozygous *MTHFR* rs1801131 C/C, under a dominant model, was associated with a decreased probability of suffering any adverse event (logOR = −0.8; 95% CI = −1.6, −0.1). Heterozygous *ADA* rs244076 G/A was also negatively associated with adverse events under a codominant model (logOR = −0.7; 95% CI = −1.2, −0.1). In addition, *ADA* rs244076 and *MTHFR* rs1801131 polymorphisms also had a significant effect on the prevalence of gastrointestinal adverse events. Patterns observed previously are corroborated: heterozygous G/A *ADA* rs244076 was associated with a reduce prevalence of gastrointestinal adverse events under codominant (logOR = −1; 95% CI = −1.9, −0.1) and overdominant (logOR = −1.1; 95% CI = −2.0, −0.2) inheritance models. Consequently, homozygous G/G is associated with having adverse events under codominant (logOR = 2.6; 95% CI = 0.7, 4.3) and recessive (logOr = 2.7; 95% CI = 0.9, 4.5) models. Moreover, as before, heterozygous *MTHFR* rs1801131 polymorphism increased the chances of suffering gastrointestinal adverse events (logOR = 0.8; 95% CI = 0.04, 1.5). Finally, homozygous A/A and G/G *FPGS* rs1544105 were negatively (logOR = −1.2; 95% CI = −2.3, −0.1) and positively (logOR = 0.7: 95% CI = 0.07, 1.3) associated with adverse events, respectively.

Sex-specific effects involved in the susceptibility to adverse events were also tested for those genes showing a significant interaction term with sex at birth. Figure 7 panels a and b show the distribution of all adverse events and gastrointestinal adverse events by sex.

Significant effects on the susceptibility of having any adverse event and gastrointestinal adverse events were found in men but not in women for *MTHFR* rs1801131 and rs1801133, *GGH* rs1800909, and *ADA* rs244076 (Figure 7). *GGH* rs1800909 heterozygous T/C was found to decrease the probability of having any adverse effect (logOR = −2.5; 95% CI = −4.4, −0.6), and the homozygous T/T was found to increase it (logOR = 2.3; 95% CI = 0.6, 4.1). *MTHFR* rs1801131 homozygous C/C was found to be associated with a decrease in both every and specifically gastrointestinal adverse events (logOR = −1.7; 95% CI = −3.4, −0.04 and logOR = −3.4; 95% CI = −5.9, −0.9; respectively). On the contrary, heterozygous C/T increases both types of adverse event (logOR = 1.9; 95% CI = 0.2, 3.8 and logOR = 4.9; 95% CI = 1.4, 8.4; respectively). Finally, *ADA* rs244076 heterozygous G/A decreases the probability of gastrointestinal adverse events (logOR = −2.1; 95% CI = −4.0, −0.1) while homozygous G/G increases the same probability (3.7; 95% CI = 0.9, 6.6).

### 3.5. Assessing Genomic Effects to Predict RA Remission and Adverse Events

Besides the estimation of the effect of different genes on RA remission and the presence of adverse events, it is also interesting to assess the contribution of these genes to a predictive model. In order to achieve this, we first chose the most relevant covariables using a lasso logistic regression. Among the 44 covariables used in this study, different sets were flagged as relevant. The lasso approach highlighted ten variables for response to MTX and AR remission: radiological erosions, tuberculosis, cardiovascular risk factors, maximum dose of MTX, duration of the MTX treatment, time between first symptoms and treatment, DAS28 levels at basal, dose of folic acid, MTX monotherapy duration, and being a smoker. Eleven variables were highlighted for the presence of any adverse events: DAS28 levels at basal, being a smoker, age, alcohol intake, obesity, tuberculosis, baldness, hypertransaminasemia, number of risk factors, maximum dose of MTX, and MTX monotherapy duration. Figure 8 shows the ROC curves and AUCs for different logistic models for AR remission and the presence of any adverse event. For each of the three dependent variables, the model with the highest AUC is the one that includes all the 44 covariables and the significant genes found in this study. Genes, by themselves, do not have predictive power (AUC = 0.56, 0.58, and 0.57 for response to MTX, AR remission, and all adverse events, respectively). Even more, it seems that adding significant genes to the lasso-flagged list of covariables does not significantly improve the AUC.

The analysis (all patients and by sex) of the response to MTX, the association with the remission of RA, the associations with the presence of any adverse event, are included in eight different complementary tables (see Appendix A).

## 4. Discussion

In the present study, we examined the impact of eight SNPs in seven genes on the effectiveness and toxicity of MTX treatment. We also included clinical records, which allowed us to consider a larger number of factors in our analysis. This may be the reason why the present results differ from those previously reported. The present results suggest that *ITPA* (rs1127354) and *ABCB1* (rs1045642) may have a pharmacogenetic effect on the efficacy of MTX measured through DAS28; *ADA* (rs244076) and *MTHFR* (rs1801133) may affects its toxicity, while *GGH* (rs1800909), *MTHFR* (rs1801131), and *FPGS* (rs1544105) may have an effect in both the efficacy and toxicity of MTX treatment.

*ITPA* gene encodes the inosine triphosphate pyrophosphatase adenosine pathway enzyme which hydrolyzes inosine triphosphate and deoxyinosine triphosphate to the monophosphate nucleotide (Figure 1). We showed that homozygote *ITPA* (rs1127354) C/C is associated with an increased probability of being a responder to MTX. This outcome was not observed in most of the previous works [7,19,20]. Also, the heterozygous (C/A) was found to be associated with non-responsiveness [26].

*FPGS* homozygous G/G genotype (rs1544105), which encodes the folylpolyglutamate synthase mitochondrial enzyme (Figure 1), has been found by this study to be associated with AR remission in males but not in females. Homozygous A/A and heterozygous A/G genotypes were also associated with an increase in gastrointestinal adverse effects in the whole dataset and in females, respectively. This outcome, in the general population, was also found in the bibliography [21]. Nevertheless, most previously published studies did not find a significant association [7,14,19,24,25,34].

*GGH* gene encodes for the gamma-glutamyl hydrolase which removes gamma-linked glutamate from MTX. Dominant homozygous T/T SNP rs1800909 was found to be associated with both levels of DAS28 (through the response to MTX and AR remission) and adverse events in men, this genotype being related to a better response to medication and fewer adverse events. Effects of *GGH* are highly heterogenous in the bibliography, with published papers which do find significant association [29] and others that find no effect whatsoever [7,28,30].

*MTHFR* gene, which encodes for the methylenetetrahydrifolatereductase enzyme, is one of the most studied genes in the context of MTX treatment. Our study found that both SNPs rs1801133 and rs1801131 are associated with MTX toxicity and rs1801131 AR remission. As can be seen in Table 1, the effects of these SNPs are not clear since it is possible to find original articles and meta-analyses with and without [13,20,24,34] significant effects reported for both efficacy and toxicity.

Lack of consistency was also found when analyzing the effects of *ABCB1* (rs1045642), which encodes the cell membrane protein P-glycoprotein. We found a significant effect of the heterozygous rs1045642 (C/T), under an overdominant model, in male responders to MTX. However, is possible to find any kind of result in the bibliography as can be observed in Table 1. Finally, heterogeneity of outcomes could also be found in *ADA* (rs244076) when comparing our outcomes with the ones at Table 1. Heterozygous G/A seems to provide the individual with protection against adverse events (Figure 6 and Figure 7); however, this is the first time *ADA* (rs244076) was found to have a significant effect on the prevalence of adverse events.

The study suffers certain limitations that need to be addressed. First, even though the sample size used in this study is among the largest for this type of study, it is still relatively limited and does not provide a representative picture of the population. This means that it may not have sufficient representation of certain genotypes, and it is possible that weaker but significant effects are not being detected. A limited sample size also prevents us from testing interactions between genes or even the development of models to assess prediction models. Furthermore, this study focused its attention on eight SNPs from seven genes, but there are more than forty genes and almost a hundred SNPs that may be involved in the pharmacokinetics of MTX. Nevertheless, our study, as far as we know, is the one that includes the greatest number of covariables and comorbidities. Our results suggest that these covariables are more relevant than genomic information when building predictive models, as shown by the ROC analysis (Figure 8).

## 5. Conclusions

In conclusion, this study has demonstrated that the pharmacogenetic analysis of MTX treatment is more complex than previously thought and that we are still far from having a complete understanding of the process. Our findings have identified an association of *ITPA* rs1127354 and *ABCB1* rs1045642 with response to MTX, polymorphisms of the *FPGS* rs1544105, *GGH* rs1800909, and *MTHFR* genes with disease remission, *GGH* rs1800909 and *MTHFR* rs1801131 polymorphisms with all adverse events, and *ADA* rs244076 and *MTHFR* rs1801131, rs1801133 polymorphisms with gastrointestinal adverse events. Findings also have highlighted the importance of considering sex-specific effects. Additionally, we have shown that the inclusion of clinical covariates is crucial for obtaining meaningful pharmacogenetic models. Further studies should take these factors into account to better understand the pharmacogenetics of MTX treatment.

## Figures and Tables

**Figure 1 pharmaceutics-15-01661-f001:**
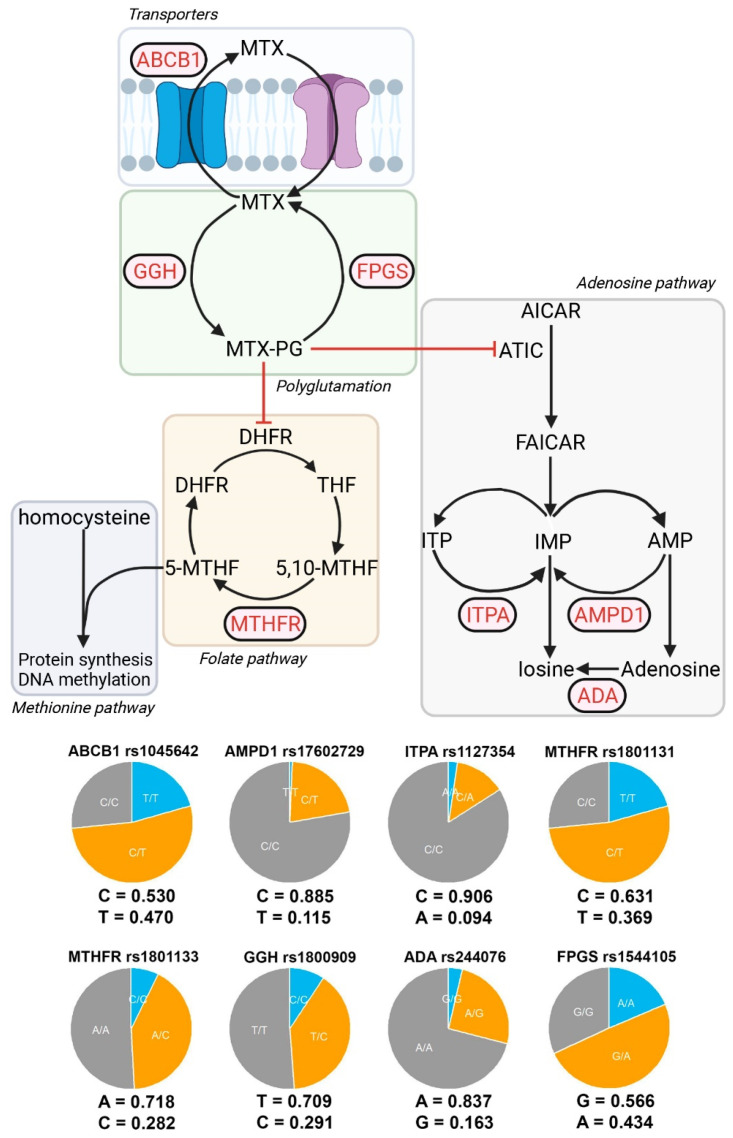
Genes and alleles under study. Partial representation of the metabolic pathways involved in the MTX effect. The genes analyzed in this study are shown in red. Eight SNP belonging to seven genes were analyzed: *MTHFR* (rs1801133 rs1801131), *ABCB1* (rs1045642), *FPGS* (rs1544105), *GGH* (rs1800909), *AMPD1* (rs17602729), *ITPA* (rs1127354), and *ADA* (rs244076). Allelic frequencies and distribution of DAS28-PCR. Allelic frequencies and genotypic pie charts are shown for each of the SNPs analyzed. DAS28-PCR distributions for each SNP and genotype are shown.

**Figure 2 pharmaceutics-15-01661-f002:**
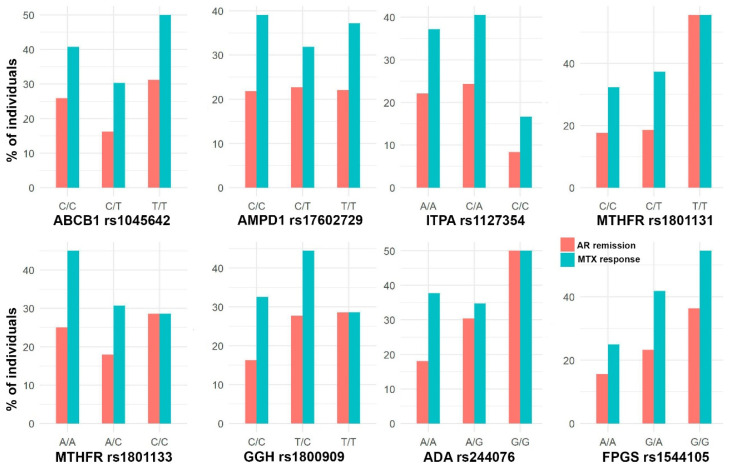
Bar plots with the percentage of individuals with AR remission (DAS28 < 2.6) and response to MTX (DAS28 < 3.2) in each SNP genotype.

**Figure 5 pharmaceutics-15-01661-f005:**
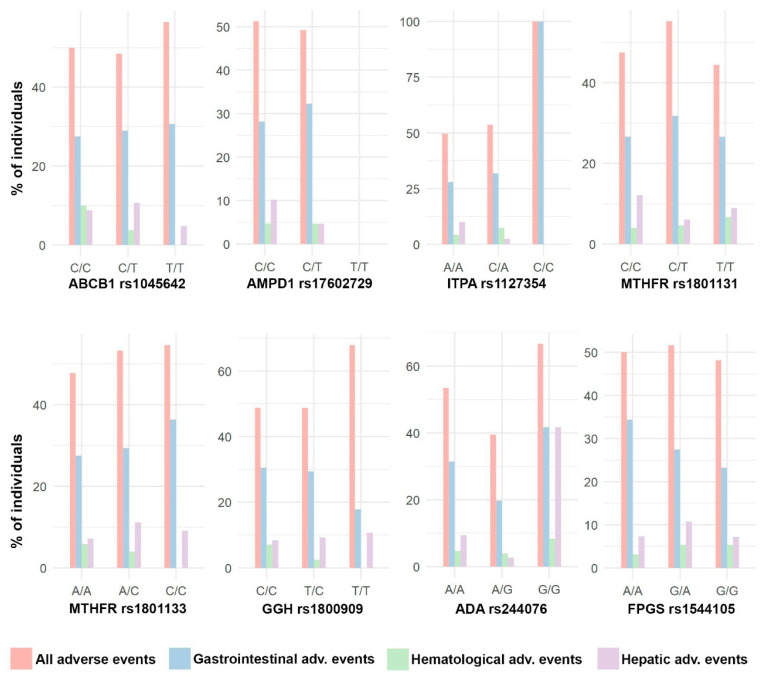
Distribution of adverse events. Bar plots with the percentage of individuals with different types of adverse events in each SNP genotype.

**Figure 6 pharmaceutics-15-01661-f006:**
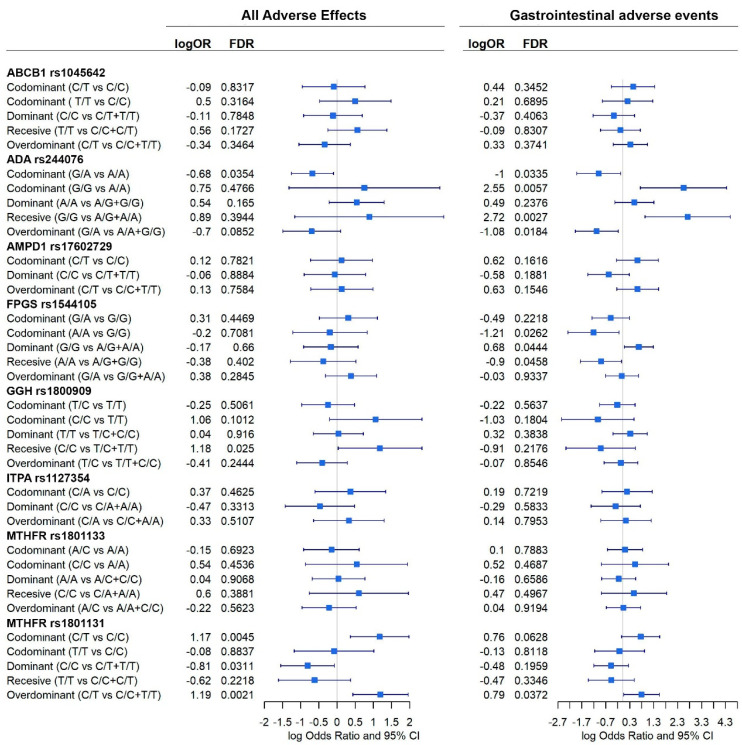
Effect of each gene, according to different inheritance models, on total adverse events and gastrointestinal adverse events. Outcomes are present in a forest plot showing the logOdds ratios with their 95% confident intervals.

**Figure 7 pharmaceutics-15-01661-f007:**
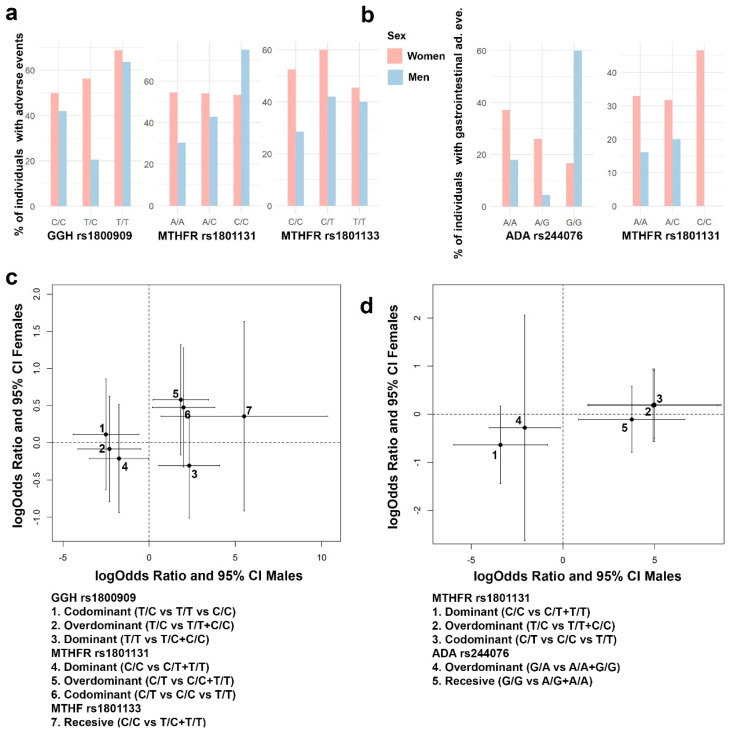
Sex-specific effects in the presence of any adverse event and gastrointestinal adverse events. (**a**) Distribution of presence of any adverse event by sex for the SNPs *GGH* rs1800909, *MTHFR* rs1801131, and *MTHFR* rs1801133. (**b**) Distribution of gastrointestinal adverse events by sex for the SNPs *ADA* rs244076 and *MTHFR* rs1801131. (**c**) Genetic *GGH* rs1800909, *MTHFR* rs1801131, and *MTHFR* rs1801133 effect estimates on having any adverse event, and 95% CI, for men (*x* axis) and women (*y* axis) are shown. (**d**) Genetic *ADA* rs244076 and *MTHFR* rs1801131 effect estimates over gastrointestinal adverse events, and 95% CI, for men (*x* axis) and women (*y* axis) are shown.

**Figure 8 pharmaceutics-15-01661-f008:**
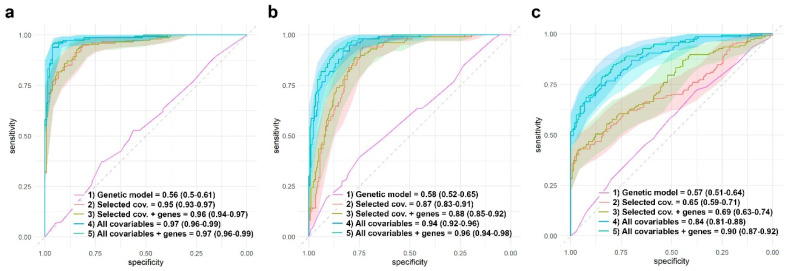
Comparison of different models AUC. Response to MTX (**a**), AR remission (**b**), and presence of any adverse event (**c**). AUC estimations were obtained through logistic regressions developed using the test group (see material and methods). (**a**) Response to MTX: ROC curves, average AUC, and 95% CI are shown for four different models: (1) AR remission vs. genes with significant effect: ABCB1 (recessive inheritance), FPGS (codominant, dominant, and overdominant inheritances), and MTHFR rs1801131 (codominant and recessive inheritances). (2) AR remission vs. 10 selected covariables by the lasso approach (radiological erosions, tuberculosis, cardiovascular risk factors, maximum dose of MTX, duration of the MTX treatment, time between first symptoms and treatment, DAS28 levels at basal, dose of folic acid, MTX monotherapy duration, and being a smoker). (3) AR remission vs. genes with significant effect + eight selected covariables by the lasso approach. (4) AR vs. all the 44 covariables. (5) AR vs. genes with significant effect + all the 44 covariables. (**b**) AR remission: ROC curves, average AUC and 95% CI are shown for four different models: (1) AR remission vs. genes with significant effect: ABCB1 (recessive inheritance), FPGS (codominant, dominant, and overdominant inheritances), and MTHFR rs1801131 (codominant and recessive inheritances). (2) AR remission vs. 10 selected covariables by the lasso approach (radiological erosions, tuberculosis, cardiovascular risk factors, maximum dose of MTX, duration of the MTX treatment, time between first symptoms and treatment, DAS28 levels at basal, dose of folic acid, MTX monotherapy duration, and being a smoker). (3) AR remission vs. genes with significant effect + eight selected covariables by the lasso approach. (4) AR vs. all the 44 covariables. (5) AR vs. genes with significant effect + all the 44 covariables. (**c**) Presence of any adverse event: ROC curves, average AUC, and 95% CI are shown for four different models. Eleven selected covariables for models (2) and (3) were: DAS28 levels at basal, being a smoker, age, alcohol intake, obesity, tuberculosis, baldness, hypertransaminasemia, number of risk factors, maximum dose of MTX, and MTX monotherapy duration. Selected genes and inheritances for models (1), (3), and (4) were ADA rs244076 (codominant inheritance), FPGS rs1544105 (dominant and codominant inheritances), GGH rs1800909 (recessive inheritance), and MTHFR rs1801131 (dominant, codominant, and overdominant inheritances).

**Table 1 pharmaceutics-15-01661-t001:** Summary of the main pharmacogenetic outcomes for methotrexate (MTX) intake in rheumatoid arthritis (RA) patients. Original studies published after 2016 were considered.

Type Study	Patients (n)	Country/ Region	Evaluation	Genes, SNP	Main Results	Ref.
Original article	45	Turkey	MTX efficacy through DAS28	*ABCB1* rs1045642	10 covariables and comorbidities included. *ABCB1* heterozygous (C/T) is associated with non-response to MTX	[16]
Original article	167	Malaysia	MTX efficacy and toxicity	*FPGS* rs10106	*ATIC* rs2372536 and rs4673993 associated with MTX adequate response in RA patients with Malay ancestry	[19]
				*GGH* rs1145078, rs3758149		
				*ATIC* rs2372536, rs4673993		
				*ITPA* rs1127354		
Original article	227	Spain	Intolerance to MTX	*RFC* rs1051266	Positive association with *GGH* AA + AG vs. GG. Inverse association with ABCC2 TT + TC vs. CC	[20]
				*GGH* rs11545078		
				*MTHFR* rs181133, rs1801131		
				*DHFR* rs1105525		
				*SHMT1* rs1979277		
				*ITPA* rs34743033		
				*ABCC2* rs717620		
				*ABCB1* rs1045642		
				*SLCOB1* rs11045879		
Original article	381	Sweden	Adverse liver effects	*MTHFR* rs1801131	*MTHFR* nominally associated but did not pass correction for clinical factors	[13]
				*TYMS* rs34743033		
				*SLCOB1* rs11045879		
Original article	330	India (Tamil)	Response to treatment (DAS28); Adverse events	*FPGS* rs1544105, rs10106	*FPGS* heterozygous rs1544105 and rs10106 associated with adverse events	[21]
				*GGH* rs11545078, rs3758149		
Original article	400	Colombia	Efficacy (DAS28 < 3.2); Toxicity	*MTHFR* rs1801133, rs1801131	*MTHFR* rs1801133 and rs1801131 associated with efficacy of MTX. *DHFR* and ATIC associated with toxicity	[14]
				*ATIC* rs2371536		
				*RFC1* rs1051266		
				*FPGS* rs1554105		
				*DHFR* rs10072026;		
Original article	100	Northern Ireland	MTX-7-OH and MTX2PG–5PG metabolite levels	*ABCC1* rs246240	*MTHFR* rs1476413 TT CC alleles associated with plasma poly-glutamate metabolites	[22]
				*MTHFR* rs1476413, rs4846051, rs17421511		
				*ABCG2* rs2231142		
				*ABCC2* rs717620, rs3740065		
				*SCLO1B1* rs4846051		
				*ABCB1* rs10280623		
				*ATIC* rs16853826		
Original article	114	Japan	Hepatotoxicity; Hepatic enzyme elevation	*ABCB1* rs1045642	Multivariable analysis corrected for different confounding variables. *ABCB1* and *ATIC* associated with MTX-induced abnormal hepatic enzyme elevation	[21]
				ABCC2 rs717620		
				*RFC1* rs1051266		
				*FPGS* rs1544105		
				*GGH* rs3758149		
				*MTHFR* rs1801131		
				*ATIC* rs16853826		
Retrospective cohort study	162	China (Han Chinese)	MTX toxicity	*MTHFR* rs1801133, rs1801131	*MTHFR* rs1801133 associated with MTX-related toxicity in East Asians. *ATIC*, *ABCB1* and *RFC1* associated in Europeans. *RFC1* associated with MTX-related toxicity in the retrospective cohort study	[7]
				*ATIC* rs2372536;		
				*ABCB1* rs1045642		
				*MTR2756* rs1805087		
				*MTRR* rs1801394		
				FPGS rs10106		
				*AMPD1* rs17602729		
				*ITPA* rs1127354		
				*GGH* rs3758149, rs1800909, rs11545078		
				*ABCC2* rs4148396		
				*ABCG2* rs2231142		
				*FPGS* rs1544105		
				*TYMS* rs34743033		
				*ADORA2A* rs2267076, rs2298383		
				*MTHFD1* rs2236225		
Meta-Analysis	16	Multiple	MTX efficacy (DAS28 < 3.2); MTX-related toxicity	*MTHFR* rs1801131, rs1801133	Associated with toxicity but not efficacy in Asian and European populations. Model corrected for folic acid us	[23]
Meta-Analysis	30	Multiple	MTX efficacy (DAS28)	34 genes 125 SNP included in the study	MTHFR rs1801131, *ATIC* rs2372536, *SCL19A1* rs2838956, and *SLC19A1* rs7499 associated with MTX efficacy	[24]
Meta-Analysis	42	Multiple	MTX toxicity	28 genes with 88 SNP included in the study	*RFC1* (rs1051266) was found to be associated with MTX toxicity	[25]
Meta-Analysis	68	Multiple	MTX efficacy (DAS28 < 3.2); MTX-related toxicity	42 genes 152 SNP included in the study	14 studies adjusted for sex and age. *MTHFR* rs1801131, *TYMS,* and *AMPD1* associated with responsiveness. *ATIC* rs2372536 and *ITPA* rs1127354 associated with non-responsiveness. *MTHFR* rs1801131, ATIC rs2372536, and *TYMS* associated with adverse events. *GGH*, *SLCO1B1* rs4149056, *SLC19A1* rs2838956 rs1131596 rs2838958 associated with absenting overall adverse events	[26]
Meta-Analysis	17	European	MTX efficacy (DAS28 < 3.2); MTX-related toxicity	*RFC1* rs1051266	*RFC1* associated with MTX efficacy in Asians	[27]
		Asian				
Meta-Analysis	16	Multiple	MTX treatment response (DAS28 > 3.2)	*ABCB1* rs1045642	*ABCB1* associated with higher drug resistance. *ABCG2* is associated with higher drug resistance	[6]
				*ABCC2* rs717620		
				*ABCC3*; *ABCC4*; *ABCC5*; *ABCG2* rs2231142		
Meta-Analysis	12	Multiple	MTX efficacy (DAS28 < 3.2); MTX-related toxicity	*ABCB1* rs1045642	*ABCB1* Associated with MTX efficacy under a recessive model. Association found mainly in Asian not in European populations	[8]
Meta-Analysis	39	European	MTX toxicity	*MTHFR* rs1801133, rs1801131	*MTHFR* rs1801133 associated with MTX-related toxicity in East Asians. *ATIC*, *ABCB1,* and *RFC1* associated in Europeans. *RFC1* associated with MTX-related toxicity in the retrospective cohort study	[7]
				*ATIC* rs2372536;		
				*ABCB1* rs1045642		
				*MTR2756* rs1805087		
				*MTRR* rs1801394		
				FPGS rs10106		
				*AMPD1* rs17602729		
				*ITPA* rs1127354		
				*GGH* rs3758149, rs1800909, rs11545078		
				*ABCC2* rs4148396		
				*ABCG2* rs2231142		
				*FPGS* rs1544105		
				*TYMS* rs34743033		
				*ADORA2A* rs2267076, rs2298383		
				*MTHFD1* rs2236225		

**Table 2 pharmaceutics-15-01661-t002:** Patient’s characteristics classified according to response to MTX and presence of any adverse effect.

		Response to MTX		RA Remission		Adverse Events	
	Individuals	DAS28 < 3.2	DAS28 > 3.2	DAS28 < 2.6	DAS28 > 2.6	YES	NO
Number	301	148	135	99	202	152	149
Demographic							
Age	49 (41–58)	53 (45–63.3)	47 (38–56)	52 (44.5–64)	48 (38.5–56)	48 (39.5–56)	52 (41–62)
Sex (Women)	202 (67.1%)	94 (63.5%)	94 (69.6)	61 (61.6%)	141 (69.8%)	113 (74.3%)	89 (59.7%)
Educational attainment							
Before high school	98 (32.5%)	44 (29.7%)	44 (32.6%)	29 (29.2%)	69 (34.1%)	46 (30.2%)	52 (34.8%)
High school	105 (34.9%)	54 (36.5%)	49 (36.3%)	33 (33.1%)	72 (35.6%)	54 (35.5%)	51 (34.3%)
University studies	98 (32.5%)	50 (33.8%)	42 (32.6%)	37 (37.4%)	61 (30.2%)	52 (34.2%)	46 (30.9%)
Activity							
Sedentary	57 (18.9%)	28 (18.9)	23 (17.1%)	19 (19.2%)	38 (18.8%)	32 (21.1%)	25 (16.8%)
Mixed Active	153 (50.8%)	71 (47.9%)	72 (53.3%)	46 (46.4%)	107 (52.9%)	79 (51.9%)	74 (49.7%)
Active	91 (30.2%)	49 (33.1%)	40 (29.6%)	34 (34.3%)	57 (28.2)	41 (26.9%)	50 (33.6%)
Lifestyle							
Smokers							
Never	118 (39.2%)	57 (38.5%)	49 (36.3%)	41 (41.4%)	77 (38.1%)	62 (40.7%)	56 (37.6%)
Ex-smoker	97 (32.2%)	63 (42.6%)	33 (24.4%)	41 (41.4%)	56 (27.8%)	46 (30.3%)	51 (34.2%)
Active smoker	86 (28.6%)	28 (18.9%)	53 (39.3%)	17 (17.2%)	69 (34.2%)	44 (28.9%)	42 (28.2%)
Alcohol							
Never	181 (60.1%)	83 (56.1%)	86 (63.7%)	55 (55.5%)	126 (62.4%)	97 (63.8%)	84 (56.4%)
Ex-alcohol drinker	33 (10.9%)	18 (12.2%)	14 (10.4%)	8 (8.1%)	25 (12.4%)	15 (7.4%)	18 (12.1%)
Active drinker	87 (28.9%)	47 (31.8%)	35 (25.6%)	36 (36.3%)	51 (25.2%)	40 (19.8%)	47 (31.5%)
Tea consumption	29 (9.6%)	11 (7.4%)	16 (11.9%)	9 (9.1%)	20 (9.9%)	17 (11.2%)	12 (8.1%)
Coffee consumption	185 (61.7%)	80 (54.1%)	76 (56.3%)	57 (57.6%)	128 (63.4%)	93 (61.2%)	92 (61.7%)
Comorbidities							
Any comorb.	234 (77.7%)	113 (76.4%)	105 (77.8%)	74 (74.7%)	160 (79.2%)	121 (79.6%)	113 (75.8%)
Cardiovascular comorb.	237 (78.7%)	32 (21.6%)	28 (20.7%)	73 (73.7%)	164 (81.2%)	123 (80.9%)	114 (76.5%)
Number of risk factors							
0	64 (21.3%)	40 (27.0%)	20 (14.8)	26 (26.3%)	38 (18.8%)	29 (19.1%)	35 (23.5%)
1	94 (31.2%)	55 (37.2%)	39 (28.9%)	41 (41.4%)	53 (26.2%)	39 (25.7%)	55 (36.9%)
2	79 (26.2%)	37 (25%)	42 (31.1%)	21 (21.2%)	58 (28.7%)	45 (29.6%)	34 (22.8%)
3	52 (17.3%)	23 (15.5%)	32 (23.7%)	10 (10.1%)	42 (20.8%)	30 (19.7%)	22 (14.8%)
4	10 (3.3%)	3 (2%)	7 (5.2%)	1 (1%)	9 (4.5%)	7 (4.6%)	3 (2.0%)
5	2 (0.6%)	0	2 (1.5%)	0	2 (0.9%)	2 (1.3%)	0
High blood pressure	114 (37.9%)	46 (31.1%)	58 (42.9%)	30 (30.3%)	84 (41.6%)	61 (40.1%)	53 (35.6%)
Diabetes Mellitus	49 (16.3%)	20 (13.5%)	29 (31.5%)	13 (13.1%)	36 (17.8%)	32 (21.1%)	17 (11.4%)
Hypercholesterolemia	137 (45.5%)	59 (39.9%)	78 (57.8%)	35 (35.4%)	102 (50.5%)	74 (48.7%)	63 (42.3%)
Obesity	72 (23.9%)	31 (20.9%)	41 (30.4%)	22 (22.2%)	50 (24.8%)	46 (30.3%)	26 (17.4%)
Ischemic heart disease	20 (6.7%)	12 (8.1%)	8 (5.9%)	8 (8.1%)	12 (5.9%)	12 (7.9%)	8 (5.4%)
Any cardiovascular pathology	63 (20.9%)	34 (22.9%)	39 (38.9%)	24 (24.2%)	39 (19.3%)	30 (19.7%)	33 (22.1%)
Vascular incidents	46 (15.3%)	19 (12.8%)	27 (20%)	17 (17.2%)	29 (14.4%)	26 (17.1%)	20 (13.4%)
Thyroid diseases	35 (11.6%)	17 (11.5%)	18 (13.3%)	11 (11.1%)	24 (11.9%)	21 (13.8%)	14 (9.4%)
Renal impairment	12 (4.0%)	5 (3.4%)	7 (5.2%)	3 (3.1%)	9 (4.5%)	9 (5.9%)	3 (2.0%)
Osteoporosis	73 (24.3%)	33 (22.3%)	31 (22.9%)	22 (22.2%)	51 (25.2%)	38 (25.0%)	35 (23.5%)
Tuberculosis	42 (14.0%)	8 (5.4%)	33 (22.9%)	4 (4.1%)	38 (18.8%)	26 (17.1%)	16 (10.7%)
Baldness	26 (8.6%)	8 (5.4%)	13 (9.6%)	3 (3.0%)	23 (11.4%)	25 (16.4%)	1 (0.7%)
Hypertransaminasemia	27 (9.0%)	12 (8.2%)	15 (11.1%)	5 (5.1%)	22 (10.9%)	27 (17.8%)	0
Depression	63 (20.9%)	25 (16.9%)	28 (20.7%)	13 (13.1%)	50 (24.8%)	39 (25.6%)	24 (16.1%)
Diseases Manifestations							
DAS28-	4.3 (3.8–5.1)	3.8 (3.6–4.1)	4.9 (4.5–5.4)	3.8 (3.5–4.2)	4.7 (4.1–5.3)	4.4 (3.9–5.1)	4.1 (3.7–4.8)
Extraarticular manifestations	71 (23.6%)	27 (18.2%)	44 (32.65)	12 (12.1%)	59 (29.2)	44 (28.9)	27 (18.1%)
Radiological erosions in hands	163 (54.1%)	65 (43.9%)	98 (72.6%)	28 (28.3%)	135 (66.8%)	88 (57.9%)	75 (50.3%)
Sjogren syndrome	33 (11.0%)	15 (10.1%)	18 (13.3)	9 (9.1%)	24 (11.9%)	18 (11.8%)	15 (10.1%)
Carpal tunnel syndrome	20 (9.9%)	7 (4.7%)	13 (9.6$)	4 (4.1%)	16 (7.9%)	11 (7.2%)	9 (6.0%)
Pulmonary affection	14 (4.7%)	5 (3.45)	9 (6.7%)	3 (3.0%)	11 (5.4%)	9 (5.9%)	5 (3.4%)
Pharmacological variables							
Use DMARD previously	81 (26.9%)	34 (22.9%)	47 (34.8%)	16 (16.2%)	65 (32.2%)	47 (30.9%)	34 (22.9%)
Number previous DMARD							
0	223 (74.1%)	124 (83.8%)	99 (73.3%)	83 (83.8%)	140 (69.3%)	108 (71.1%)	115 (77.2%)
1	54 (17.9%)	21 (14.2%)	33 (24.4%)	13 (13.1%)	41 (20.3%)	29 (19.1%)	25 (16.8%)
2	16 (5.3%)	8 (5.4%)	8 (5.9%)	3 (3.0%)	13 (6.4%)	9 (5.9%)	7 (4.7%)
3	6 (2%)	0	6 (4.4%)	0	6 (3.0%)	4 (2.6%)	2 (1.3%)
4	2 (0.7%)	0	2 (1.5%)	0	2 (1%)	2 (1.3%)	0
Time between diagnosis and treatment with DMARD (days)	0 (0–3)	0 (0–2)	1 (0–3)	0 (0–2.1)	1 (0–9)	0 (0–4)	0 (0–3)
Initial dose of prednisona (mg/day)	10 (5–10)	5 (5–10)	10 (5–10)	5 (5–10)	10 (5–10)	10 (5–10	10 (5–10
Dose of folic acid (mg/week)	5 (5–5)	5 (5–5)	5 (5–10)	5 (5–5)	5 (5–10)	5 (5–5)	5 (5–10)
Maximum dose of MTX (mg/week)	15 (15–20)	15 (15–20)	20 (15–20)	15 (12.5–15)	17.5 (15–20)	15 (15–20)	15 (15–20)
Maximum tolerated dose of MTX (mg/week)	15 (15–20)	15 (12.5–17.5)	15 (15–20)	15 (12.5–15)	17.5 (15–20)	15 (15–20)	15 (15–20)
Time between diagnosis and treatment with MTX (days)	1 (0–15)	0 (0–7.2)	2 (0–14)	0 (0–4)	2 (0–28.7)	1 (0–22.5)	1 (0–9)
MTX monotherapy duration (months)	51 (20–101)	86 (43–125)	24 (10–60)	84 (38–118.5)	42 (15–88)	47 (14.7–96.2)	59 (27–103)
Antibodies							
Rheumatoid Factor							
Positive	242 (80.4%)	111 (75%)	131 (97%)	78 (78.9%)	164 (81.2%)	120 (78.9%)	122 (81.9%)
Mean Value (DE)	61.6 (21–176)	51.2 (20.2–125)	75 (30.1–198)	44.9 (16.1–105.3)	85.3 (25.6–202)	63.8 (19.2–207.6)	60 (25–159)
Antipeptide citrullinate (ACPA)	193 (64.1%)	91 (61.5%)	102 (75.6%)	59 (59.6%)	134 (66.3%)	97 (63.8%)	96 (64.4%)
Antinuclear antibodies	27 (9%)	9 (6.1%)	18 (13.3%)	4 (4.0%)	23 (11.2%)	14 (9.2%)	13 (8.7%)

Statistics: Individual characteristics were summarized using standard descriptive statistics: median (25 and 75% quartiles) for continuous variables and count (percentage) for categorical variables.

**Table 3 pharmaceutics-15-01661-t003:** Description and frequencies of the adverse effects in the group of patients treated with MTX (n = 301).

Description of Adverse Effect	N	%
Adverse effects (total)	152	50.50
Haematologics	14	4.65
Leukopenia (alone or as part of pancytopenia)	9	2.99
Thrombocytopenia (alone or as part of pancytopenia)	12	3.99
Pancytopenia	7	2.33
Hepatic (elevated transaminases)	27	8.97
2 × ULN	13	4.32
3 × ULN	8	2.66
>3 × ULN	6	1.99
Gastrointestinal	87	28.90
Nausea, vomiting, fullness	80	26.58
Diarrhea	10	3.32
Neurological/general	55	18.27
Alopecia	26	8.64
Cutaneous	13	4.32
Rash	8	2.66
Others	5	1.66
Oral ulcers	20	6.64
Accelerated nodulosis	6	1.99
Interstitial involvement	2	0.66

*ULN*: upper limit of normal for transaminases.

## Data Availability

The data presented in this study are available in deidentified form on request from the corresponding author. The data are not publicly available due to privacy restrictions.

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
