# Peer review of "Pharmacogenetic Sex-Specific Effects of Methotrexate Response in Patients with Rheumatoid Arthritis"

_pharmaceutics, 2023, doi:10.3390/pharmaceutics15061661_

Round 1

Reviewer 1 Report

This manuscript investigate the association between 8 polymorphisms of genes encoding proteins involved in the metabolism and transport of MTX, in a mixed group of RA patients treated with MTX.

I have major issues with the study.

The concept of studying response to MTX it difficult to undertake. Due to the heterogeneity of RA and the very different type of response to MTX that can be obtained based on the disease course, notwithstanding the change in clinical practice about HOW and WHEN to prescribed MTX, added to the change in the use of classification criteria to diagnose RA since 2010, all make this study unacceptable in the current form.

Response to MTX will vary based on a number of factor that will evolved with time hence not universal and that will change the course of disease.

·         Time at which MTX is prescribed in the course of disease (hence within the window of opportunity for it to induce remission or not, being DMATDs naïve or not etc…)

·         Length of symptom duration before MTX is prescribed

·         Previous drugs used before MTX => use of oral steroids, NSAIDs, others DMARDs which is also country specific and will not allow to translate finding across borders.

·         Drugs prescribed WITH MTX =>  mono therapy, combined therapy with other synthetic or even biologic DMARDs, use of the treat to target approach to rationalise treatment protocols.

·         Time allowed on MTX will condition WHETHER the drugs has had a chance to work

There is no description in the method of WHEN the response was determines, hence preventing any of the data proposed to be sufficiently informed to be acceptable.  

The study should limit itself to

Ø  Early RA (i.e. homogenous duration of symptom limited to 12 or 24 months)

Ø  DMARDs naïve (allowing for steroid and NSAIDs as current practice in Spain indeed)

Ø  Monotherapy and/or adherence to treat-to-target protocol should be considered notably as remission is an outcome considered by the authors.

Ø  Standardised time for considering whether MTX is successful or not

There is major discrepancies between data displayed in figures with only 1 data point below DAS28=3 and data presented in Tables with 99 patients with DAS28<2.6 (remission)

Minor:

1.       Table 1 is irrelevant to the study in the current format (=> supplementary material) while a table focussing on the 8 SNPs used in the study and describing all finding related to these specifically, would have brought a lot more rational to the work in line with Figure 1.  What does MA abbreviates for ? not described in legend.

2.       I have never heard to DAS28-PCR ? what does it stand for ? is it some kind of a cumulative DAS28 variable ?

3.       I have no idea what time between diagnosis and treatment with “FAME “ means ?

4.       Time between diagnosis and treatment is irrelevant as diagnosis can be made anytime, depending on when the patients come/has access to clinic. It is duration of symptoms until diagnosis that is relevant.  This is estimated by the patients and by the clinician and is needed to establish the diagnosis hence it should be available as a variable.

Author Response

We would like to express our gratitude to reviewer 1 for their valuable comments and feedback. We have carefully considered the major issue raised by the reviewer regarding the difficulty in assessing response to MTX due to the heterogeneity of RA and the response to MTX. We have changed the manuscript accordingly to address its issues.

We would like to emphasize that in our manuscript, we have presented one of the most comprehensive assessments of MTX pharmacology that has been published to date. We have employed a total of 44 covariates to account for all possible sources of heterogeneity and have been able to recruit a prospective cohort for our study. We have added two more covariables to the previous version of the analysis: DAS28 levels at a basal stage, previous to treatment and the time between first symptoms and treatment.

We acknowledge that the response to MTX can vary greatly among RA patients. To address this issue, we used different variables. Firstly, DAS28-CRP at a basal stage which, in combination with the time from symptoms to treatment and other variables, allowed us to control for differences at basal stage among the participants. Secondly, we used different variables like rheumatic factor antibodies, anti-citrullinated peptide antibodies, and antinuclear antibodies, to provide additional information on disease status.

We have indeed addressed all reviewer concerns in the model as we develop below:

  1. Time at which MTX is prescribed in the course of the disease: In our manuscript, we have taken this into consideration by incorporating three different covariates. First, we included the time between diagnosis and treatment with DMARD to account for the overall disease duration. Second, we added the time between diagnosis and treatment with MTX to specifically address the timing of MTX prescription. Finally, we included MTX monotherapy duration to assess the impact of treatment duration on response. These covariates have not been commonly reported in MTX pharmacogenetic studies and we believe they add valuable information to our analysis.
  2. We have added length of the symptom duration before MTX is prescribed as the reviewer asked for.
  3. Drugs prescribed with MTX. While the main focus of our study is the pharmacogenetics of MTX monotherapy, we recognize the importance of these additional factors and their potential impact on MTX response. To address this issue, we included in the model the effects of previous disease-modifying antirheumatic drugs, folic acid, and predisnona. Also, we introduce the maximum dose of MTX as a control variable.

In summary, we have added numerous covariates to assess the pharmacogenomics effect of MTX treatment, including 44 control variables. This approach is one of the most comprehensive to date and has enabled us to draw robust conclusions. Nonetheless, replication of these findings in future studies is warranted.

The reviewer suggested that there is no description in the method of when the response was determined. While it is true that the patient sampling was done in chronological order, it is important to clarify that the determination of the response was not based on the order of patient sampling. The response was determined based on the clinical outcomes of patients after they were treated with MTX. Furthermore, as we mentioned earlier, we have included several covariates in our model to account for potential sources of heterogeneity, including the time between diagnosis and treatment with MTX, length of the symptom duration before MTX and the duration of MTX monotherapy. These covariates should help to mitigate any potential biases arising from differences in the timing of response determination among patients.

Finally, the reviewer suggested that the study should be limited to early RA, DMARDs without prior treatment, monotherapy and/or adherence to the treatment protocol to the target should be considered in particular, and the standardized time to consider whether MTX is successful or not. No. We appreciate the comment, but we consider that the approach of the study, in this case, is appropriate, although this approach that it proposes is interesting to take into account in future research. We consider that, in the present study, limiting the selection of individuals to only those with homogeneous symptoms for 12 or 24 months could introduce arbitrary bias in our analysis and we did not include different time variables to account for this heterogeneity. On the other hand, we have already accounted for the use of DMARDs and treatment time, and we believe that evaluating individuals on MTX monotherapy allows us to better assess the pharmacogenomics of MTX. In addition, we use already validated approaches to standardize the measure of MTX success, such as the DAS28-CRP, which consists of evaluating response to MTX (DAS28<3.2) and RA remission (DAS28<2.6). This approach is widely used in the literature and has been validated in several articles.

We greatly appreciate your feedback and hope this clarification addresses your concerns.

We consider that your contribution has been crucial for the improvement of the manuscript, and we thank you again.

Reviewer 2 Report

1. A new table showed the percentage of different adverse events is needed.

2. Patients with RA often received many medication. How did the GI adverse events define. For example, NSAIDs may cause the adverse effect of nausea, vomiting, and abdominal pain in patients with RA.

3.For the DMARDs used, why some patients received 3-4 previous DMARDs? The inclusion criteria “currently receiving MTX monotherapy at the time of inclusion” is OK. “who had received MTX treatment at any time during the course of their disease” did these patients receive combination therapy. Once these patients received combination therapy, the efficacy could be due to the effect of other DMARDs.

4. For the side effect, many medication except MTX could cause elevated liver function. How did the PI define the liver adverse effect is due to MTX?

5. How to define the RA status of disease activity is related to MTX+steroid usage. Eg. used MTX+steroid for a duration then measured the RA disease activity.

6.Line 142: antipeptide citrullinated → should change to anti-cyclic citrullinated peptide antibody; anti-CCP.

7.In the abstract, the authors should briefly described the name of genes SNP that associated with MTX efficacy and its effect on remission status.

8.The label of figures is wrong. I did not found figure 6-9, as mentioned in the text.

9. The results of uni-variate and multi-variate logistic regression should be present in tables.

10.What does DAS28-PCR mean? The authors used DAS28-ESR or DAS-CRP for the classification of RA disease activity?

Author Response

We appreciate the comments, those made, which improve the quality of the manuscript. We will answer them below.

  1. A new table showed the percentage of different adverse events is needed.

#1. According to reviewer suggestion, a new table (Table 3) about different adverse effects present in the study population has been included.

  1. Patients with RA often received many medications. How did the GI adverse events define. For example, NSAIDs may cause the adverse effect of nausea, vomiting, and abdominal pain in patients with RA.

#2. According to appearance of any significant MTX-related adverse event at any time during treatment, the following episodes were included in the definition of toxicity, provided they could not be explained by other causes:

  • Hematological involvement: anemia (Hb<9), leukopenia (<4,000), thrombopenia (<150,000)
  • Elevation of transaminases (GOT and/or GPT) above 2 times the upper limit of normality.
  • Gastrointestinal symptoms (nausea, vomiting, diarrhea, abdominal pain, sensation of abdominal fullness, anorexia) in connection with weekly MTX administration
  • General symptoms: non-specific discomfort related to the administration (fatigue, malaise, arthralgia, myalgia and/or fever) and non-specific and reversible neurological symptoms (lethargy, lightheadedness, drowsiness, dizziness, instability, disorientation).
  • Diffuse alopecia
  • Mucositis
  • Dermatitis
  • Accelerated rheumatoid nodulosis
  • Acute interstitial pneumonitis

Furthermore, in the revised manuscript, it has been included a new Table (Table 3) with the percentage of all adverse effects in the group of patients treated with MTX.

  1. For the DMARDs used, why some patients received 3-4 previous DMARDs? The inclusion criteria “currently receiving MTX monotherapy at the time of inclusion” is OK. “who had received MTX treatment at any time during the course of their disease” did these patients receive combination therapy. Once these patients received combination therapy, the efficacy could be due to the effect of other DMARDs.

#3. Thank you very much for the comment. Patients with a longer disease evolution or who had previously been treated in other centres could have initially received salazopyrin, azathioprine, gold salts (aurotiomalate), leflunomide, etc. And they went on to receive MTX monotherapy due to lack of response or intolerance to previous DMARDs.

Inclusion criteria: Receiving, or having received at some time, treatment with MTX monotherapy. During monotherapy, patients could also take concomitant nonsteroidal anti-inflammatory drugs, oral prednisone, and folic acid, but not other DMARDs from baseline.

No patients receiving MTX in combination with other DMARDs were included.

One of the exclusion criteria was "having received biologic agents, cyclophosphamide, or MTX combination therapy prior to monotherapy."

Therefore, In the 2.1. Participant Recruitment subsection, we are including two paragraphs with this information Page 7, lines 90-99):

In inclusion criteria: “…receiving, or have ever received, treatment with methotrexate monotherapy. During monotherapy, patients could also take concomitant nonsteroidal anti-inflammatory drugs, oral prednisone, and folic acid, but not other DMARDs from baseline.”

In exclusion criteria: "Furthermore, patients who had received biologics, cyclophosphamide, or combination of MTX with other DMARDs prior to monotherapy were excluded.

  1. For the side effect, many medications except MTX could cause elevated liver function. How did the PI define the liver adverse effect is due to MTX?

#4. The following episodes were included in the definition of toxicity, provided they could not be explained by other causes: elevation of transaminases (GOT and/or GPT) above 2 times the upper limit of normality.

Other causes were ruled out by withdrawing suspected drugs or when there was a clear temporal relationship with starting or escalating MTX, adding folic acid and temporarily interrupting MTX.

  1. How to define the RA status of disease activity is related to MTX+steroid usage. Eg. used MTX+steroid for a duration then measured the RA disease activity.

#5. We started with an evaluation of the activity (DAS28PCR) prior to starting MTX (baseline) and the evaluation was repeated every three months. Patients were generally taking low-dose prednisone before starting MTX and when the response was favourable, the prednisone dose was gradually decreased. Once achieved, the response to MTX had to be maintained for more than 6 months.

Patients were classified as responders to MTX if they reached a state of low clinical activity, defined by a DAS28-PCR value <3.2, and as non-responders when they remained with a value ≥3.2.

To establish response criteria, patients had to have received MTX monotherapy at the highest tolerated dose for at least 6 months. On the other hand, cases in which MTX was discontinued at any time due to intolerance or toxicity were considered non-responders. On the other hand, responders who achieved and maintained a DAS28-CRP <2.6 were considered to be in clinical remission, regardless of whether they continued MTX monotherapy or no treatment.

According with reviewer comment, we are included a paragraph in 2.3. Variables of Interest subsection (page 7, lines 145-148): “To establish response criteria, patients had to have received MTX monotherapy at the highest tolerated dose for at least 6 months and responders who achieved and maintained a DAS28-CRP <2.6 were in clinical remission, regardless of whether they continued MTX monotherapy or no treatment”.

6.Line 142: antipeptide citrullinated → should change to anti-cyclic citrullinated peptide antibody; anti-CCP.

#6. We have change this accordingly.

7.In the abstract, the authors should briefly describe the name of genes SNP that associated with MTX efficacy and its effect on remission status.

#7. We have changed the manuscript accordingly.

  1. The label of figures is wrong. I did not found figure 6-9, as mentioned in the text.

#8. We have corrected this issue.

  1. The results of univariate and multi-variate logistic regression should be present in tables.

#9. In our analysis we did not develop any multi-variate models (several independent variables) but multivariable logistic models (several dependent variables). When performing multivariable general linear models, it is important NOT to perform simple models with each of the dependent variables. This is called “table 2 fallacy” by many guidelines like PROBAST (Prediction model Risk Of Bias Assessment Tool) our statements like TRIPOD which aims to improve the transparency of the reporting of a prediction model study regardless of the study methods used.

Nevertheless, following the reviewer guide, we have added a xlsx file with supplementary results. In those results it is possible to find the outcomes for the different multivariable logistic models.

10.What does DAS28-PCR mean? The authors used DAS28-ESR or DAS-CRP for the classification of RA disease activity?

#10. We thank the reviewer for pointing out this mistake. In this study we used DAS28-CRP for the classification of RA disease activity. We have updated the text.

Reviewer 3 Report

Minor comments:

Line 44: The authors should mention the dosages used for treatment of hämatologie diseases compared to those used for treatment of RA .

Table 1: What does this mean that type study is "osteoarthritis"? The type of the study as well as the control group with study size should be clarified in separate rows, and separated from the meta-analyses possibly including the same studies (abbreviation of MA is not listed in the legend)

Figure 1: why partial presentation and not overview of main mechanisms involved ...?

Line 345: MTX should always be abbreviated, please check.

Line 388 and others: English language should be improved.

Major comment:

The "Data availability statement" should be added.

Author Response

We appreciate the comments, those made, which improve the quality of the manuscript. We will answer them below.

Minor comments:

  1. Line 44: The authors should mention the dosages used for treatment of haematologic diseases compared to those used for treatment of RA.

#1. According to reviewer suggestion, this phrase has been changed to: “MTX is also used in the treatment of other autoimmune diseases (such as psoriasis, 10 to 25 mg/week) and certain types of cancer (such as lymphoblastic leukemia, 200 mg/m2 intravenous over two hours up to high doses of 1–3 g/m2 as a 24-hour continuous infusion), usually a somewhat higher dose than for RA (7.5 to 20 mg/week)

  1. Table 1: What does this mean that type study is "osteoarthritis"?. The type of the study as well as the control group with study size should be clarified in separate rows, and separated from the meta-analyses possibly including the same studies (abbreviation of MA is not listed in the legend)

#2. In Table 1, in column “Type study” OA is not “osteoarthritis” but “original article”, and MA is “meta-analysis”. Instead, the format of Table 1 we presented was limited by the page setup. Anyway, we have modified the Table 1 according to the reviewer's suggestions.

  1. Figure 1: why partial presentation and not overview of main mechanisms involved?

#3. Figure 1 is an illustration where the main pathways where the enzymes with the greatest scientific evidence participate in the response (efficacy and toxicity) to MTX are represented, mainly the steps that occur within the target cell, not adding the processes that occur into intestinal, renal or hepatic cells, steps related to the absorption and elimination of MTX. We did not include the entire MTX mechanism as we thought it could cause confusion.

  1. Line 345: MTX should always be abbreviated, please check.

#4. In this Line there are not “methotrexate”. In any case, according to the reviewer's suggestion, all the text has been checked.

  1. Line 388 and others: English language should be improved.

#5. Thank you very much for the remark. We have revised and improved the English of that sentence and of the entire document.

Major comment:

  1. The "Data availability statement" should be added.

#6. The “Data availability statement” has been included (See lines 470-472): “Data Availability Statement: The data presented in this study are available in deidentified form on request from the corresponding author. The data are not publicly available due to privacy restrictions.”

Reviewer 4 Report

Comments to Author:

I think the aim of the study is very important since MTX is the anchor drug for patients with RA. To make the report more useful for readers, the authors are kindly asked to check the points below.

1.             It seems that the patients were recruited since 1990 however, 2010 ACR/EULAR RA classification criteria were applied. How about the inconsistency?

2.             What does ‘DAS28-PCR’ (Page 6, line 39) mean?

3.             Table 2: Was MTX administered orally or subcutaneously?

Author Response

Thank you very much for your comments, which will help to improve the manuscript. We will answer them below.

  1. It seems that the patients were recruited since 1990 however, 2010 ACR/EULAR RA classification criteria were applied. How about the inconsistency?

#1. Thank you very much for your comment. In order to participate in the study, patients had to meet the following inclusion criteria:

  • > 18 years old.
  • Diagnosed with RA according to the 1987 ACR criteria.
  • Protocolized follow-up in the RA monographic consultation (this information is on the patient's evaluation sheet).
  • Receiving, or have ever received, treatment with methotrexate monotherapy. During monotherapy, patients could also take concomitant nonsteroidal anti-inflammatory drugs, oral prednisone, and folic acid, but not other DMARDs from baseline.

On the other hand, a consecutive sampling was used as the patients arrived at the consultation and participation was proposed to all patients who were being followed up in RA monographic consultations between 2012 and 2015.

Of the patients who signed the informed consent, those who also met the ACR/EULAR 2010 criteria were included.

We have added an explanatory sentence in Material and Methods section (Page 5, lines 88-99)

  1. What does ‘DAS28-PCR’ (Page 6, line 39) mean?

#2. The “Disease Activity Score” (DAS) is a scoring system to assess the activity of rheumatoid disease, having been recommended by the “European League Against Rheumatism” (EULAR) for this purpose both in clinical studies and in practice daily clinic. The DAS index for measuring disease activity was developed during the 1980s (Van Riel et al., 1983) and was published by Van der Heijde et al. (1990). DAS uses 44 joints in its count. The DAS index combines information regarding the number of swollen, tender, and acute phase reactant joints, and a global measure of health status. DAS28 is a simplified version using only 28 joints for counting and was published by Prevvo et al. (1995).

Therefore, DAS28-PCR is a variant of DAS28 that substitutes the erythrocyte sedimentation rate for the C-reactive protein count and takes 28 specific joints.

Formula: DAS28-CRP=0.56* √(TJC)+0.28* √(SJC)+0.36*1n(CRP+1)*0.014* (GDAP)+0.96

TJC= Tender Joint Count; SJC= Swollen Joint Count; CRP= C-reactive protein; GDAP= global health

We have added an explanatory sentence in Material and Methods section (Page 7, lines 135-144)

References:

- Van Riel PL, Reekers P, van de Putte LB, Gribnau FW. Association of HLA antigens, toxic reactions and therapeutic response to auranofin and aurothioglucose in patients with rheumatoid arthritis. Tissue Antigens. 1983 Sep;22(3):194-9. doi: 10.1111/j.1399-0039.1983.tb01191.x.

- Van der Heijde DM, van 't Hof MA, van Riel PL, Theunisse LA, Lubberts EW, van Leeuwen MA, van Rijswijk MH, van de Putte LB. Judging disease activity in clinical practice in rheumatoid arthritis: first step in the development of a disease activity score. Ann Rheum Dis. 1990 Nov;49(11):916-20. doi: 10.1136/ard.49.11.916.

- Prevoo ML, van 't Hof MA, Kuper HH, van Leeuwen MA, van de Putte LB, van Riel PL. Modified disease activity scores that include twenty-eight-joint counts. Development and validation in a prospective longitudinal study of patients with rheumatoid arthritis. Arthritis Rheum. 1995 Jan;38(1):44-8. doi: 10.1002/art.1780380107.

  1. Table 2: Was MTX administered orally or subcutaneously?

#3. MTX was administered under a rapid escalation pattern, initially administered orally in a single dose up to 10mg/day, in two doses starting at 10mg/day. They were switched to the subcutaneous route if they did not tolerate oral administration well, the amount of 15 mg per day was exceeded or the desired efficacy was not achieved.

We have added an explanatory sentence in Material and Methods section (Page 5, lines 102-105)

Round 2

Reviewer 1 Report

nA

Reviewer 2 Report

Thank you for the responses. The study is very informative for rheumatologist.